# Vertebral Heart Scale for the Brittany Spaniel: Breed-Specific Range and Its Correlation with Heart Disease Assessed by Clinical and Echocardiographic Findings

**DOI:** 10.3390/vetsci8120300

**Published:** 2021-12-01

**Authors:** Anthony Kallassy, Elodie Calendrier, Nora Bouhsina, Marion Fusellier

**Affiliations:** National Veterinary School of Nantes, Oniris, Atlanpole, La Chantrerie, 44300 Nantes, France; kallass@hotmail.com (A.K.); elodie.calendrier@yahoo.fr (E.C.); nora.bouhsina@oniris-nantes.fr (N.B.)

**Keywords:** canine, VHS vertebral heart score, vertebral heart size

## Abstract

The vertebral heart scale (VHS) was proposed by Buchanan and Bucheler as an objective method for estimating heart size in dogs. However, several studies have reported significant variation between breeds. The purpose of this retrospective study was to evaluate the VHS and to suggest a useful upper limit for normal heart size in Brittany Spaniels. The VHS was measured using a right lateral view in twenty-eight normal dogs and fifteen dogs with myxomatous mitral valve disease. The mean ± SD (standard deviation) VHS was 10.6 ± 0.2 vertebrae (v) in the normal dogs, which differs significantly from the mean VHS of 9.7 ± 0.5 v in Buchanan’s original study with dogs of various breeds. The VHS in the dogs with myxomatous mitral valve disease was 11.9 ± 1.1 v. With a threshold value of 11.1 vertebrae, the sensitivity, specificity, positive and negative predictive values for diagnosing a cardiomegaly are 90%, 72%, 53% and 96%, respectively.

## 1. Introduction

Despite the advent of echocardiography, thoracic radiography remains a frequently used tool for the evaluation of heart size in veterinary medicine. It is also an integral part of the management of cardiac disease, as the criteria for heart enlargement used for the American College of Veterinary Internal Medicine (ACVIM) classification for myxomatous mitral valve disease, including the breed-adjusted radiographic vertebral score [1,2].

Traditionally, methods using planimetry and cardiothoracic ratio have been described to evaluate heart size with the aim of maximizing the accuracy of radiographic diagnosis [2,3]. However, the use of these methods has been undermined by the lack of precise measurement points, by the marked variations in thoracic conformation that occur in dogs, as well as by the variations in the appearance of the cardiac silhouette that may result from the phase of respiration, and concurrent factors such as non-cardiac disease, weight, phase of the cardiac cycle, etc. [4,5,6,7].

The vertebral heart score (VHS) measurement has been reported to be a more objective method than a traditionally used method for assessing cardiomegaly in dogs. Devised by Buchanan and Bucheler, the method consists of measuring the long and short axes of the cardiac silhouette on lateral radiographs and comparing the sum of these measurements with the thoracic vertebral length beginning at T4. Based on the evaluation of 100 normal dogs representing a variety of breeds, a reference range of 9.7 ± 0.5 vertebral bodies was established. As the VHS was <10.5 in 98% of the dogs, this was suggested to be a clinically useful upper limit for normal heart size in most breeds [4].

The VHS is considered to be easy to perform, and it produces results independent of the thoracic dimensions, gender, or left or right recumbency. It has also proven to be a reliable and reproducible method to calculate the size of the cardiac silhouette independently of observer experience. Unlike the traditional heart size assessment methods, it has the advantage of well-defined measurement points and an objective numerical measurement [8,9,10]. Another study determined a significant correlation between VHS and the echocardiographic parameters used to monitor the degree of cardiomegaly [6].

However, one limitation of the method may be the use of a normal range based on a group of 100 dogs of various breeds. Since the original report, several studies have shown significant breed-specific differences [7,8,9,10,11], with a mean VHS higher than 10.5 in Cavalier King Charles Spaniel, Labrador Retriever, Boxer [12], Pug, Pomeranian, Bulldog, Boston Terrier [11], retired racing Greyhounds [13], Whippet [14], and Norwich Terriers [15].

Therefore, it would be more accurate to use breed-specific ranges, particularly for inexperienced observers who may be especially prone to false-positive interpretations when examining radiographs. This would improve accurate radiographic interpretation of the cardiac silhouette, which has a direct impact on the medical management of specific conditions [16].

The Brittany Spaniel is a common hunting breed in France. It is ranked 15th among the most popular breeds in 2017 according to the LOF (livre des origines françaises) registration statistics [17]. There are no available studies regarding the prevalence of heart disease in this breed, although a recent retrospective study showed that 6% of the mortality in this breed is due to a cardiovascular disease [18]. Furthermore, Brittany Spaniels is believed by French cardiologists and radiologists to have a larger cardiac silhouette than other breeds on thoracic radiographs. Yet, there have been no published papers confirming this fact, which, hence, motivated us to conduct this study.

The aims of this study were to determine the reference range for the VHS in Brittany Spaniels, to evaluate whether the normal VHS value proposed by Buchanan and Bucheler (9.7 ± 0.5) is applicable, and to determine whether clinical signs of cardiac disease, gender, and body condition scores have any influence on vertebral heart scale measurements in this breed. We hypothesized that a normal Brittany Spaniel would have a higher mean VHS than the original references.

## 2. Materials and Methods

### 2.1. Animals

Fifty-three owned animals were recruited retrospectively. A search of the medical records at Oniris Hospital of mature Brittany Spaniels that had 2 to 4 thoracic radiography images taken between 1 September 2007 and 30 June 2018 was conducted. Dogs without a heart murmur and clinical signs indicative of heart failure were selected as a control group. Any dog with non-sinus arrhythmia, tachypnoea, dyspnoea, syncope, exercise intolerance, ascites, or cyanosis, were excluded from the control group. The dogs that had an auscultable murmur and cardiac disease detected by echocardiography were selected as the cardiac disease group. The medical records of selected dogs were reviewed to extract the age at the radiographic examination, gender, body weight, body condition score (1 to 9) when available, clinical evaluation, thoracic radiographs, and CBC (complete blood count), urinalysis, and biochemical panel when available. In cases where echocardiography was performed, the results were also recorded. All the animals with a condition that could affect the size of the heart, such as dehydration or animals under perfusion, were excluded from the study.

### 2.2. Thoracic Radiography

All Brittany Spaniels that had at least one lateral and one ventrodorsal or dorsoventral projection performed between 1 September 2007 and 30 June 2018 were selected for the study.

To be included in the study, all of the radiographs had to be of good technical and diagnostic quality (density, contrast, sharpness). They were taken at the point of full inspiration, and care was taken to avoid any rotation of the body as this could influence the shape and the size of the cardiac silhouette [19]. Any patient with vertebral deformities, or with a condition leading to inaccurate measurements were excluded from the study. All the radiographs were performed on non-sedated animals using a digital radiography system (X-ray generator PICKER, Uniontown, OH, USA, Computed radiography, Profect, FUJI, Tokyo, Japan) with a grid and a focus to film distance of 100 cm.

Measurements were made using an adjustable calliper in millimetres, which was then converted to vertebral units (v) exact to 0.1 v, according to the method described by Buchanan and Bucheler [4]. All of the measurements were performed by the same examiner (EC) using DICOM PACS view radiography computer software (Synapse, Fuji, Tokyo, Japan). In the right lateral radiograph, the long axis of the heart was measured from the ventral border of the left main stem bronchus to the most distant contour of the cardiac apex (L). The maximal short-axis of the heart was measured perpendicular to the long-axis (S). The two measures were repositioned over thoracic vertebrae beginning with the cranial edge of T4, using an adjustable calliper (Figure 1 and Figure 2).

### 2.3. Echocardiography

All of the echocardiographic examinations were performed by the successive imaging assistants of the hospital, under supervision of the head of the imaging department (M.F.), using an Esaote Mylab70XVG device (Esaote SpA, Genoa, Italy). The operators were blinded to the radiographic results. All echocardiograms were performed on un-sedated dogs in the standing position [20]. The following measurements were each taken over at least 3 cardiac cycles, and the mean was recorded as follows: the end-diastolic LA/Ao ratio obtained from the right parasternal short-axis 2D (cut-off value 1.2) [21], the left ventricular internal diameter at end-diastole (LVIDd), and left ventricular internal diameter at end-systole (LVIDs) measured on the M-mode echocardiogram, obtained from the right parasternal short-axis view [22,23]. M-mode values were used to derive the fractional shortening (FS%). Normalized dimensions were calculated according to the following formulae: normalized LVIDd (LVIDDN) = LVIDd(cm)/(BW (kg))^0.294^ (cut-off value 1.7) [24]. Mitral valve disease was diagnosed when thickening and incomplete apposition of the valve leaflets during systole were observed, with secondary mitral valve regurgitation. The cardiac dogs were then classified into 2 groups according to the results of the echocardiography. All dogs with structural mitral valve disease with no echocardiographic evidence of cardiac remodelling formed group 1. Dogs with more advanced mitral valve disease with echocardiographic findings of left atrial dilation with or without left ventricular dilation formed group 2 [25].

### 2.4. Statistical Analysis

The statistical analyses were performed using a computerized statistical software package (R 3.5.0^®^ software, Vienna, Austria). For all of the VHS measurements, the mean and the standard deviation (SD) were calculated, and differences with *p* < 0.05 were considered significant. After Shapiro–Wilk test, a non-parametric Wilcoxon-Mann–Whitney test was used to determine whether there were significant differences in the VHS values in right lateral recumbency in various subcategories. It was used to compare the VHS in the control group to the cardiac disease group with cardiac remodelling (group 2), and to compare the cardiac diseased group 1 to group 2. VHS of Brittany Spaniels, with a body condition score ≤ 5, were compared to those with a body condition score ≥ 6 by mean of a Wilcoxon-Mann–Whitney test.

## 3. Results

Fifty-three dogs (37 females and 16 males) were recruited. Thirty-one dogs were included in the control group and twenty-two were included in the cardiac disease group as they suffered from a cardiac disease.

Not all of the thoracic radiographs were of good technical and diagnostic quality. In total, 28 and 15 right lateral view, and 11 and 3 left lateral view radiographs were selected in the control and cardiac disease group, respectively. Thus, right lateral radiographs were used for the VHS calculations.

The mean age in our sample was 9.2 ± 4.17 years (2 to 17), 7.9 ± 3.79 years in the healthy group (5.5 to 11.5) and 11.9 ± 3.65 in the cardiac disease group (8.2 to 15.6). The mean weight was 16.4 kg ± 3.73 (10 to 24), the body condition score was available for 26 dogs, with a mean of 5.9 ± 3.1 (4 to 8).

In the cardiac disease group, all dogs had myxomatous mitral valve disease, 12 of which had concurrent tricuspid valve disease, 11 had left atrial enlargement, 5 also had an increased left ventricular internal diameter in diastole.

VHS and the body condition score

There were no significant differences in the VHS values between different body score conditions (*p* = 0.3306).

VHS of the control group versus the cardiac disease group

There were significant differences between the value of the VHS for the control group and the cardiac disease group 2 (*p*-value < 0.05). The mean value for the right lateral VHS was significantly higher in the cardiac disease group 2 compared to the control group with a score of 11.9 ± 1.1 v (range, 11.4 to 13.4) versus 10.6 ± 0.2 v (range, 10.4 to 10.8), respectively. The median right lateral VHS in the control and cardiac disease groups were 10.6 and 11.9, respectively (Figure 3).

Sensitivity, specificity, and positive and negative predictive values of the VHS in Brittany Spaniel dogs

True positives are cardiac diseased Brittany Spaniels with a VHS higher than the high value of the reference range. True negatives are normal dogs with a VHS less than the reference range. With a threshold value of 11.1 vertebrae, the sensitivity, specificity, and positive and negative predictive values were 90% (95% C.I. 62–98), 72% (95% C.I. 55–84), 53% (95% C.I. 38–66) and 96% (95% C.I. 90–100), respectively. The Youden’s index is estimated at 0.63.

## 4. Discussion

The aims of the present study were to determine whether the mean VHS of 9.7 ± 0.5 vertebrae on lateral radiographs as determined by Buchanan and Bucheler is applicable for Brittany Spaniel, and to set a threshold value between a normal dog and one with myxomatous mitral valve disease. The mean VHS on twenty-eight right lateral radiographs was 10.6 ± 0.2 v. The normal range encompassing 95% of the values was hence estimated to be 10.4 to 10.8. It is larger than the reference mean VHS of 9.7 ± 0.5 v of Buchanan. Only 46% of the VHS values in the present study were below the upper limit of the normal 10.5 v suggested by Buchanan and Bucheler. Interbreed variations of the VHS have been described in a large variety of canine breeds. Reports have shown that the VHS is higher than 10.5 in Cavalier King Charles Spaniel (10.6 ± 0.5), Labrador Retriever (10.8 ± 0.6), Boxer (11.6 ± 0.8) [12], Pug (10.7 ± 0.9), Pomeranian (10.5 ± 0.9), Bulldog (12.7 ± 1.7), Boston Terrier (11.7 ± 1.4) [11], retired racing Greyhounds (10.5 ± 0.1) [13], Whippet (11.3 ± 0.5) [14], and Norwich Terriers(10.6 ± 0.6) [15]. We supposed that this high value of VHS obtained in Brittany Spaniel that are medium-sized dogs comes from the selection of cobby silhouettes with solid body, short and powerful and strong bone structure (Standard FCI N°95 /05.05.2003 /F-EPAGNEUL BRETON-ORIGINE: France).

Buchanan and Bucheler did not detect any significant differences in the VHS from radiographs taken in right or left recumbency [4]. On the other hand, Bavegems et al. [14] and Kraetschmer et al. [26] reported that the right lateral recumbent VHS values were 0.3 v higher than the left lateral recumbent VHS values. In the present study, the VHS was measured only on right lateral radiographs. Therefore, when interpreting VHS values in Brittany Spaniels, right recumbency is preferable.

This study determined that the VHS in right lateral recumbency in Brittany Spaniels is not related to body condition scores. Although it is in accordance with other studies [11], our results are in contrast with the study of Norwich Terriers that showed a significantly greater VHS with a body condition score ≥ 6 than those with a body condition ≤ 5 [15].

In the present study, female dogs were overrepresented (70% versus 30% of males) and the mean age of the normal dog group was higher than in similar studies (7.9 years versus 5.4 years for Whippets [14], 5.7 years in Bulldogs [11]). This is explained by the high number of thoracic radiographs carried out for bitches that had undergone mammary tumour staging. No comparison between male and female was then performed.

Several reports have shown that the type of cardiac disease can influence the accuracy of a diagnosis based on measurement of the VHS. The scale appears to be inaccurate in dogs suffering from cardiac diseases associated with concentric hypertrophy or dysrhythmias [12]. On the other hand, the scale was most accurate for cardiac diagnosis in the Yorkshire Terrier and the Cavalier King Charles breeds affected mainly by mitral insufficiency, which is more likely to be discernible by radiography [12].

In the present study, all of the dogs in the cardiac disease group were diagnosed with a myxomatous mitral disease, which is the most common cardiac condition in dogs. Males are slightly overrepresented (9 males versus 6 females) and the mean age was higher than the normal group (11.9 versus 7.9 years). This is consistent with the data in the literature, since the prevalence of myxomatous valve disease is higher in elderly dogs and the disease is approximately 1.5 times more common in males than in females [27]. Based on this study, it appears that values of the VHS lower than 11.1 for the Brittany Spaniel should provide approximately 90% sensitivity and 72% specificity for ruling out cardiomegaly related to myxomatous mitral valve disease.

The measurements were made by a single operator to avoid operator related bias, even though the variation in the mean VHS is minor between different observers. The operator was a veterinary student in their final year with a low level of experience. This should, however, not interfere with the results of the study because this individual had been trained to identify anatomical landmarks for correct measurements of the VHS. According to Hansson et al., the VHS method is independent of observer experience but dependent on the individual observer’s selection of reference points and the transformation of long- and short-axis dimensions into VHS units [5].

## 5. Conclusions

In conclusion, findings indicate that Brittany Spaniels have a mean VHS larger than the values reported by Buchanan and Bucheler in the general canine population. It is, therefore, important to take the breed into account and to standardize the radiographic methods when evaluating heart size in thoracic radiographs to avoid over-interpretation of cardiac enlargement. This study showed that a VHS greater than 11.1 v on right lateral recumbency should be considered to be a clear indication of cardiac enlargement. Nevertheless, while the VHS method provides useful indicators of heart enlargement, it cannot be used as the only mean of diagnosing cardiac disease. This method can be completed with VLAS (Vertebral Left Atrial Size) measurement when echocardiography is unavailable [28,29,30,31].

In other cases, further clinical investigation such as echocardiography is appropriate to probe for the presence of underlying heart disease.

## Figures and Tables

**Figure 1 vetsci-08-00300-f001:**
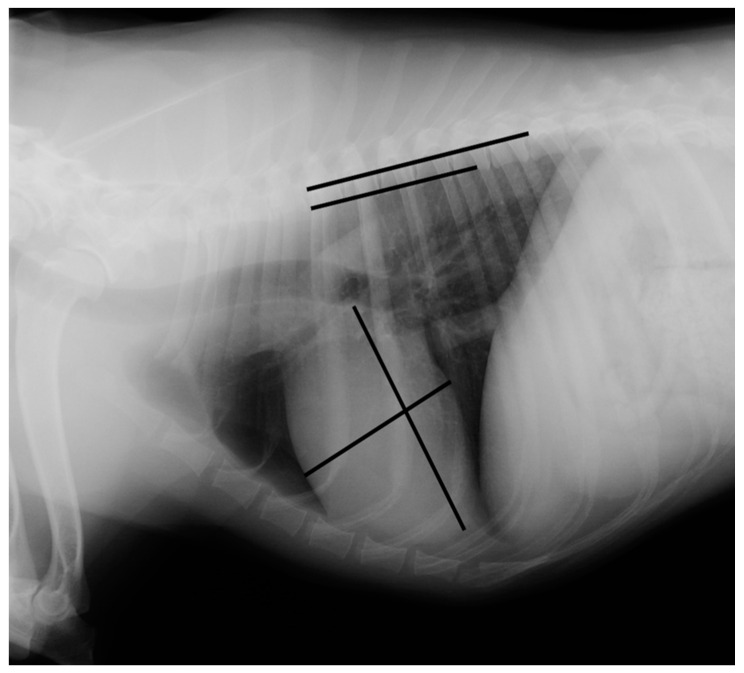
Measurement of the vertebral heart scale in a right lateral radiograph illustrating an example of a vertebral heart scale in a Brittany Spaniel of 10.8 v (the image was acquired using a PICKER CONVIX 80–UNIVERSIX 120 device (Picker International, Uniontown, OH, USA); the kVp, mA, and time settings were not recorded). Two lines are drawn on the heart to measure its long and short axes. They are then transposed onto the spine and recorded as the number of vertebrae beginning with the cranial edge of T4. These values are estimated to one decimal place and added to obtain the vertebral heart size.

**Figure 2 vetsci-08-00300-f002:**
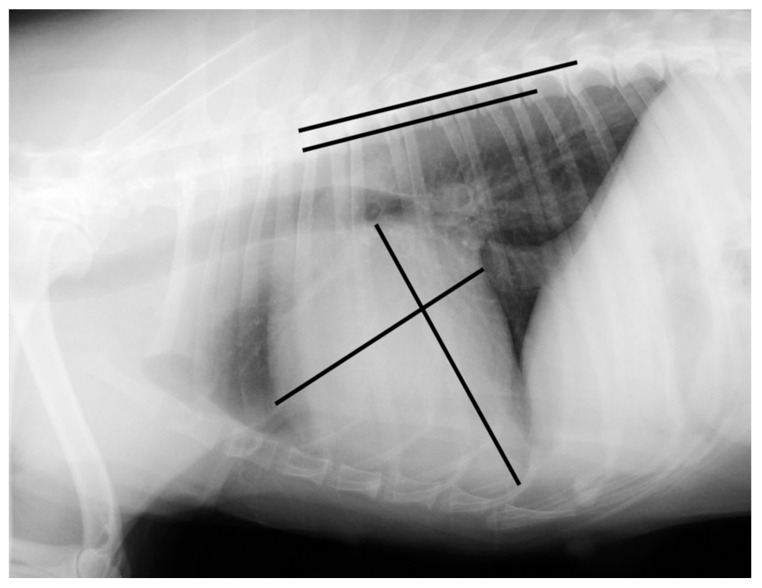
Right lateral radiograph illustrating an example of a vertebral heart scale calculation in a Brittany Spaniel suffering from a myxomatous mitral valve disease with a vertebral heart scale of 11.6 v (the image was acquired using a PICKER CONVIX 80–UNIVERSIX 120 device (Picker International, Uniontown, OH, USA); the kVp, mA, and time settings were not recorded). Two lines are drawn on the heart to measure its long and short axes. They are then transposed onto the spine and recorded as the number of vertebrae beginning with the cranial edge of T4. These values are estimated to one decimal place and added to obtain the vertebral heart size.

**Figure 3 vetsci-08-00300-f003:**
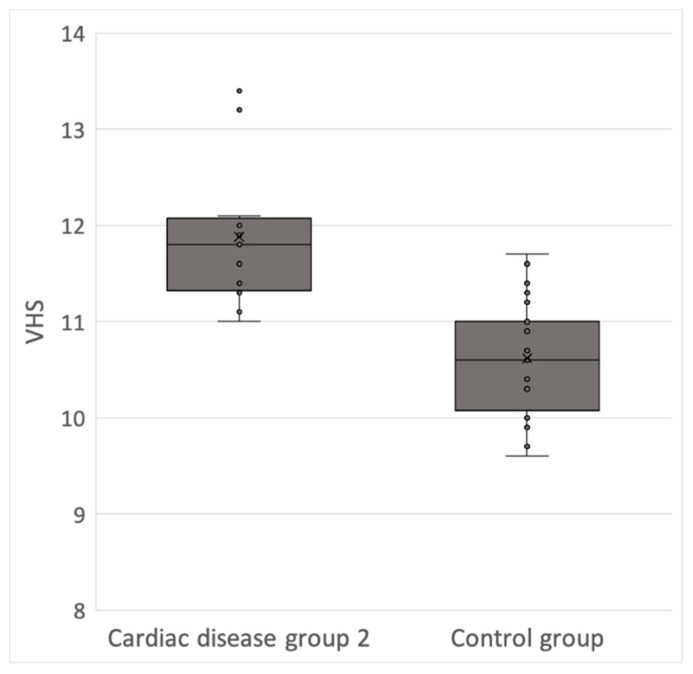
Box plots illustrating VHS values for the two groups (Control group vs. Cardiac disease group 2). Each dot corresponds to the VHS value of a patient.

## Data Availability

Not applicable.

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
