# Peer review of "Vertebral Heart Scale for the Brittany Spaniel: Breed-Specific Range and Its Correlation with Heart Disease Assessed by Clinical and Echocardiographic Findings"

_vetsci, 2021, doi:10.3390/vetsci8120300_

Round 1

Reviewer 1 Report

Overall comments:

First of all, I congratulate the authors for their paper. I realize the great effort needed to perform a clinical study.

The article is a similar to other articles enumerate in the bibliography section. But it is interesting.

Nowadays, the VHS limits used for normal dogs are 9.7 to 10.5 (includes the 98% of normal dogs), and 10.5 is the more used cutoff for cardiomegaly. This value: 10.5 is close to the VHS described in this study.

On the other hand, the authors said that all the cardiac-group dog had mitral valve disease, but just 7 dogs had left atrial enlargement and just 3 dogs had ventricular enlargement. This means that 8 cardiac dogs must have a normal size heart, so it is difficult to understand the clear difference between both groups: normal and cardiac dog.  I suggest dividing cardiac dogs according to ACVIM guidelines (B1, B2, C, D). We must keep in mind that B1 dogs have a normal VHS.

Moreover, I miss discussion of the VHS obtain in this particular breed, with respect to other breeds. The VHS obtained are quite similar to the VHS in Cavalier King Charles Spaniel and Norwich Terriers. Have the authors some idea to explain these similarities in spite of different body weights between those breeds?

I assume that the conformation of the vertebrae is normal for each animal in this breed, but this could be clarified due to the effect of abnormal shaped vertebrae or other spine anomalies in this specific measurement.

Specific comments:

Line 91-92: The author said the radiographs had to be of good technical and diagnostic quality. I suggest including cites that explain which is a good technique and quality for the radiographs.

 Line 111: There are at least two different ways to measured left atrium to aortic ratio using the right parasternal short-axis view, please clarify which one the authors have chosen for this study.

Line 113: I suggest including cites at the end of the paragraph to clarify the echocardiography methods used.

Line 114: I’m not an expert in statistics, but T-test are used with normally data or a big amount of data.  In this study the number is quite low. Have the authors calculated the type of distribution? If the answer is yes, please clarified.

Line 170: I find rare the paragraph: 3.2. figures and tables. I suggest eliminating it. On the other hand, the results presented in table 1 (except for the n) are showed through the text, so maybe redundant. The figures are mentioned through the text so they do not need a separated paragraph.

Line 200-205: In my opinion this paragraph may be a limitation of the study nor a subject of the discussion section.

Line 208: The authors must clarify that there are just 2 male dogs in their study. Maybe this asseveration is not realistic.

Line 219-221: It is necessary to include cites at the end of this assertion.

Line 232: keep in mind the overall comment related to cardiac group dogs.

Conclusion:

In my opinion the article needs some corrections including methodology.

Author Response

REVIEWER 1

Overall comments: First of all, I congratulate the authors for their paper. I realize the great effort needed to perform a clinical study.The article is a similar to other articles enumerate in the bibliography section. But it is interesting. Nowadays, the VHS limits used for normal dogs are 9.7 to 10.5 (includes the 98% of normal dogs), and 10.5 is the more used cutoff for cardiomegaly. This value: 10.5 is close to the VHS described in this study. On the other hand, the authors said that all the cardiac-group dog had mitral valve disease, but just 7 dogs had left atrial enlargement and just 3 dogs had ventricular enlargement. This means that 8 cardiac dogs must have a normal size heart, so it is difficult to understand the clear difference between both groups: normal and cardiac dog.  I suggest dividing cardiac dogs according to ACVIM guidelines (B1, B2, C, D). We must keep in mind that B1 dogs have a normal VHS.

The authors agree with reviewer’s comment and have modified the modifying data dividing cardiac dogs in two groups “All dogs with structural mitral valve disease with no echocardiographic evidence of cardiac remodeling formed the group 1. Dogs with more advanced mitral valve disease with echocardiographic findings of left atrial with or without left ventricular dilation formed the group 2 [23]3

Moreover, I miss discussion of the VHS obtain in this particular breed, with respect to other breeds. The VHS obtained are quite similar to the VHS in Cavalier King Charles Spaniel and Norwich Terriers. Have the authors some idea to explain these similarities in spite of different body weights between those breeds?

The authors thank the reviewer for his question and we have added few lines of comment in the discussion “We supposed that this high value of VHS obtained in Brittany Spaniel that are medium-sized dogs comes from the selection of cobby silhouettes with solid body, short and powerful and strong bone structure (Standard FCI N°95 /05.05.2003 /F - EPAGNEUL BRETON - ORIGINE : France).”

I assume that the conformation of the vertebrae is normal for each animal in this breed, but this could be clarified due to the effect of abnormal shaped vertebrae or other spine anomalies in this specific measurement.

The authors agree with this comment and we have added a line in the material and methods section : “Any patient with vertebral deformities, or with a condition leading to inaccurate measurements were excluded from the study.” 

Specific comments:

Line 91-92: The author said the radiographs had to be of good technical and diagnostic quality. I suggest including cites that explain which is a good technique and quality for the radiographs.

We added a reference “To be included in the study, all of the radiographs had to be of good technical and diagnostic quality (density, contrast, sharpness). They were taken at the point of full inspiration, and care was taken to avoid any rotation of the body as this could influence the shape and the size of the cardiac silhouette [17] »

 Line 111: There are at least two different ways to measured left atrium to aortic ratio using the right parasternal short-axis view, please clarify which one the authors have chosen for this study.

Line 113: I suggest including cites at the end of the paragraph to clarify the echocardiography methods used.

Thanks to the suggestion of the reviewer, the authors clarified the echocardiographic methods by adding a paragraph “All echocardiograms were performed on unsedated dogs in the standing position [19]. The following measurements were each taken over at least 3 cardiac cycles, and the mean was recorded as follows: the end-diastolic LA/Ao ratio obtained from the right parasternal short‐axis 2D [20], the left ventricular internal diameter at end‐diastole (LVIDd), and left ventricular internal diameter at end‐systole (LVIDs) measured on the M‐mode echocardiogram, obtained from the right parasternal short‐axis view [18,21]. M‐mode values were used to derive the fractional shortening (FS%). Normalized dimensions were calculated according to the following formulae: normalized LVIDd (LVIDDN) = LVIDd(cm)/(BW (kg))0.294; normalized LVIDs (LVIDSN) = LVIDs(cm)/(BW(kg)) [22]. Mitral valve disease was diagnosed when thickening and incomplete apposition of the valve leaflets during systole were observed, with secondary mitral valve regurgitation. The cardiac dogs were then classified into 2 groups according to the results of the echocardiography. All dogs with structural mitral valve disease with no echocardiographic evidence of cardiac remodeling formed the group 1. Dogs with more advanced mitral valve disease with echocardiographic findings of left atrial with or without left ventricular dilation formed the group 2 [23] »

Line 114: I’m not an expert in statistics, but T-test are used with normally data or a big amount of data.  In this study the number is quite low. Have the authors calculated the type of distribution? If the answer is yes, please clarified.

Thanks to the suggestion of the reviewer, the authors clarified the statistic tests and modified the tests using a non parametric Wilcoxon-Mann-Whitney test. The modification have been made in the manuscript.

Line 170: I find rare the paragraph: 3.2. figures and tables. I suggest eliminating it. On the other hand, the results presented in table 1 (except for the n) are showed through the text, so maybe redundant. The figures are mentioned through the text so they do not need a separated paragraph.

The authors agree with this comment and deleted table 1.

Line 200-205: In my opinion this paragraph may be a limitation of the study nor a subject of the discussion section.

The authors move this paragraph toward a limitation section.

Line 208: The authors must clarify that there are just 2 male dogs in their study. Maybe this asseveration is not realistic.

These difference between the two groups has already been explained in the discussion “This is explained by the high number of thoracic radiographs carried out for bitches that had undergone mammary tumor staging »

Line 219-221: It is necessary to include cites at the end of this assertion.

The authors thank the reviewer for this comment and have added a citation “The scale appears to be inaccurate in dogs suffering from cardiac diseases associated with concentric hypertrophy or dysrhythmias [10] »

Line 232: keep in mind the overall comment related to cardiac group dogs.

Done, thank you for your reviewing.

Conclusion:

In my opinion the article needs some corrections including methodology.

Reviewer 2 Report

Reviewer‘s comments

Vertebral heart scale for the Brittany Spaniel: breed-specific range and its correlation with heart disease assessed by clinical and echocardiographic findings

In general:

This is an interesting study, especially with the aspect that different breeds have different VHS “normal” values and that it is necessary to have as much informations as possible for different breeds, like the Brittany Spaniel. In the statistical and results part of this study are some confusing aspects, which must be corrected or analyzed in a different way. The group of dogs with a mitral valve disease are not divided in normal size and dogs with heart enlargement diagnosed by echocardiography. There are no data available for the echocardiographic classification in normal and enlarged and which part of the heart is then enlarged, because this would be an interesting aspect for the VHS measurement and the sensitivity and specificity. It would be great if the authors could add these missing data.

Abstract:

The abstract must be adjusted to the corrected data

Here are written the 28 and 15animals definitively used.

Introduction:

The introduction is well written with an adequate number of citations.

Aims of the study are well described.

Material and methods:

Animals

Please specify some aspects (see later in the results part)

Stastical analysis:

Line 116: The mean and standard deviation is used, are these data checked for normality? Which test was used for normality analysis?

Line 121: In this sentence a prospectively selected group of dogs is described. This is new, please specify this group in the “animals” paragraph.

Please check all used data for a normal distribution.

Results:

I am sorry, but you have to correct this results part, because in the first paragraph (starting with line 130) you describe the whole population and calculate mean of age, body weight… That is fine

Then you have 31 dogs in the control group and 22 in the disease group

After that you included only the dogs with right lateral radiographs and good quality.

From this point you only talk about this new selected dogs with 28 and 15.

It is not written how many dogs are male and female of this population, how old they are…

  • It would be necessary to calculate this data new for the “end” population (28 and 15) you used for the radiographic interpretation

Line 148:

This is a statistical calculation, which is not reliable, because you only have 2 male dogs. You describe a mean value for this 2 dogs, this is statistically not correct.

Line 165:

Please make a description for the diseased dogs and how many of this showed an increase in cardiac dimensions, because a diseased dog with a mitral valve insufficiency can have a normal cardiac dimension. So the calculation diseased versus normal is in this way not reliable. You have to correct this analysis.

In the results part it is necessary to describe which parts of the heart were echocardiographically enlarged and what happened with the VHS, because this is the essential point.

Discussion:

Line 190: Not only to find dogs with DMVD, ore specifically dogs with DMVD and cardiomegaly.

Line 206: The gender effect must be discussed more intensively because there were only these 2 male dogs

Line 219: It is correct, that dogs with a DMVD are good candidates for a VHS  measurement, but again it is necessary to describe the real sizes of the heart measured in echo and classify the dogs in B1, B2…and discuss it than again.

Author Response

Reviewer‘s commentsVertebral heart scale for the Brittany Spaniel: breed-specific range and its correlation with heart disease assessed by clinical and echocardiographic findings

In general: This is an interesting study, especially with the aspect that different breeds have different VHS “normal” values and that it is necessary to have as much informations as possible for different breeds, like the Brittany Spaniel. In the statistical and results part of this study are some confusing aspects, which must be corrected or analyzed in a different way. The group of dogs with a mitral valve disease are not divided in normal size and dogs with heart enlargement diagnosed by echocardiography. There are no data available for the echocardiographic classification in normal and enlarged and which part of the heart is then enlarged, because this would be an interesting aspect for the VHS measurement and the sensitivity and specificity. It would be great if the authors could add these missing data.

The author would like to thank the reviewer for all his comments. Some corrections have been made to clarify the methodology particularly echocardiographic method and statistical analysis. Two groups of cardiac dogs have been separated with a group without heart enlargement and a second with heart enlargement.

Abstract:

The abstract must be adjusted to the corrected data

The authors thank the reviewer for his comment. The abstract contains the 28 normal and 15 cardiac animals definitively used.

Introduction:

The introduction is well written with an adequate number of citations.

Aims of the study are well described.

The authors thank the reviewer for his comment.

 Material and methods:

Animals

Please specify some aspects (see later in the results part)

The authors have clarified highlighted aspects.

Stastical analysis:

Line 116: The mean and standard deviation is used, are these data checked for normality? Which test was used for normality analysis?

Normality has not been checked. A non-parametric Wilcoxon Mann Whitney test has been used to compare the different results. It is not necessary to certify normality to perform this test.” A non-parametric Wilcoxon-Mann-Whitney test was used to determine whether there were significant differences in the VHS values in right lateral recumbency in various subcategories. It was used to compare the VHS in the control group to the cardiac disease group with cardiac remodeling (group 2), and to compare cardiac diseased group 1 to group 2. Linear regression analysis was used to assess the correlation between the body score and the VHS. VHS of Brittany Spaniels with a body condition score 5 were compared to those with a body condition score  6 by mean of a Wilcoxon-Mann-Whitney test »

Line 121: In this sentence a prospectively selected group of dogs is described. This is new, please specify this group in the “animals” paragraph.

The authors thank the reviewer and apologize for this mistake. There is no prospectively selected group and we corrected this section.

Please check all used data for a normal distribution.

A non-parametric Wilcoxon Mann Whitney test has been used to compare the different results. It is not necessary to certify normality to perform this test

Results:

I am sorry, but you have to correct this results part, because in the first paragraph (starting with line 130) you describe the whole population and calculate mean of age, body weight… That is fine

Then you have 31 dogs in the control group and 22 in the disease group

After that you included only the dogs with right lateral radiographs and good quality.

From this point you only talk about this new selected dogs with 28 and 15.

It is not written how many dogs are male and female of this population, how old they are…

  • It would be necessary to calculate this data new for the “end” population (28 and 15) you used for the radiographic interpretation

The authors thank the reviewer for this comment. This part has been corrected as suggested: “Not all of the thoracic radiographs were of good technical and diagnostic quality. In total, twenty-eight and fifteen right lateral view, and eleven and three left lateral view radiographs were selected in the control and cardiac disease group respectively. Thus, right lateral radiographs were used for the VHS calculations.The mean age in our sample was 9.2 years (2 to 17), 7.9 years in the healthy group (5.5 to 11.5) and 11.9 in the cardiac disease group (8.2 to 15.6). The mean weight was 16.4 kg (10 to 24), the body condition score was available for 26 dogs, with a mean of 5.9 (4 to 8).”

Line 148:

This is a statistical calculation, which is not reliable, because you only have 2 male dogs. You describe a mean value for this 2 dogs, this is statistically not correct.

This part has been deleted as suggested.

Line 165:

Please make a description for the diseased dogs and how many of this showed an increase in cardiac dimensions, because a diseased dog with a mitral valve insufficiency can have a normal cardiac dimension. So the calculation diseased versus normal is in this way not reliable. You have to correct this analysis.

In the results part it is necessary to describe which parts of the heart were echocardiographically enlarged and what happened with the VHS, because this is the essential point.

As suggested by the reviewer the author modified the materiel and methgod section and resultat section as follow : All echocardiograms were performed on unsedated dogs in the standing position [20]. The following measurements were each taken over at least 3 cardiac cycles, and the mean was recorded as follows: the end-diastolic LA/Ao ratio obtained from the right parasternal short‐axis 2D [21], the left ventricular internal diameter at end‐diastole (LVIDd), and left ventricular internal diameter at end‐systole (LVIDs) measured on the M‐mode echocardiogram, obtained from the right parasternal short‐axis view [22,23]. M‐mode values were used to derive the fractional shortening (FS%). Normalized dimensions were calculated according to the following formulae: normalized LVIDd (LVIDDN) = LVIDd(cm)/(BW (kg)) 0.294; normalized LVIDs (LVIDSN) = LVIDs(cm)/(BW(kg)) [24]. Mitral valve disease was diagnosed when thickening and incomplete apposition of the valve leaflets during systole were observed, with secondary mitral valve regurgitation. The cardiac dogs were then classified into 2 groups according to the results of the echocardiography. All dogs with structural mitral valve disease with no echocardiographic evidence of cardiac remodeling formed the group 1. Dogs with more advanced mitral valve disease with echocardiographic findings of left atrial dilation with or without left ventricular dilation formed the group 2 [25]. » « There were significant differences between the value of the VHS for the control group and the cardiac disease group 2 (p-value < 0.05). The mean value for the right lateral VHS was significantly higher in the cardiac disease group 2 compared to the control group with a score of 11.9 ± 1.1v (range, 11.4 to 13.4) versus 10.6 ± 0.2v (range, 10.4 to 10.8) respectively. The median right lateral VHS in the control and cardiac disease groups were 10.6 and 11.9, respectively.

Discussion:

Line 190: Not only to find dogs with DMVD, ore specifically dogs with DMVD and cardiomegaly.

This sentence has been corrected “…to set a threshold value between a normal dog and one with myxomatous mitral valve disease and left atrial dilation.”

Line 206: The gender effect must be discussed more intensively because there were only these 2 male dogs

This paragraph has been modified and statistical analysis has been removed.” In the present study, female dogs were overrepresented (70% versus 30% of males) and the mean age of the normal dog group was higher than in similar studies (8.2 7.9 years versus 5.4 years for Whippets [14], 5.7 years in Bulldogs [11]). This is explained by the high number of thoracic radiographs carried out for bitches that had undergone mammary tumor staging. No comparison between male and female was then performed.“

Line 219: It is correct, that dogs with a DMVD are good candidates for a VHS measurement, but again it is necessary to describe the real sizes of the heart measured in echo and classify the dogs in B1, B2…and discuss it than again.

Groups have been considered to separate cardiac dogs without heart enlargement and cardiac dogs with heart enlargement as detailed above.

Reviewer 3 Report

This study could potentially provide clinicians important information on VHS values in healthy Brittanny Spaniels. However, the methods used to conduct the study are not adequate and the statistics are inappropriate. I suggest the authors to correct the weaknesses of their work and re-submit it  to the journal in future.

Line 78: you say that “Fifty-three owned animals were recruited retrospectively”. However, in line 121 you say “..those selected prospectively…” You should specify how many dogs were selected retrospectively and how many were included prospectively.

Lines 80-81: I believe that the absence of murmur and the absence of clinical signs of heart failure are insufficient criteria to rule out cardiac disease. How were the ECG and echocardiography results of these dogs? How many of them have had an echocardiographic examination and how many have not? What do you mean by "clinical signs indicative of heart failure“. It should be specified. Moreover, also general physical examination, CBC, urinary analysis and biochemical panel are essential tools to assess the clinical status of the dogs and to rule out dehydration or other conditions that can affect heart size

Lines 85-86: how many dogs undergone echocardiography? it should be specified.

Lines 91-92: What does it mean “all of the radiographs had to be of good technical and 91 diagnostic quality “ it should be specified.

Lines 107-108: It should be specified who performed the echocardiographic examinations.

Lines 111-113: Usually, in small animals echocardiography, the Left Atrium to aortic ratio is performed in early ventricular diastole (Rishniw 2000, Hansson 2002). Why did you measure it at end diastole? Can you provide references about this method? What echocardiographic reference intervals did you consider normal for the left atrium and left ventricle?

Statistical analysis:  Did you assess the distribution of the collected variables? The Student t-test is not suitable to compare more than two groups. Authors should use an adequate statistical test to compare more than two groups with each other. Linear regression should be used to assess the relationship between independent and dependent variables, not the correlation between two variables.

Lines 130-133: the measure of central tendency “mean” should be expressed together with SD.

Lines 140-142: What criteria did you use to diagnose mitral valve disease? You should specify this in M&M

What reference intervals did you use to diagnose left atrial enlargement? You should specify this in M&M

What reference intervals did you use to diagnose left ventricular enlargement? You should specify this in M&M

Lines 143-147: I believe that it is not correct to statistically compare the results obtained from two different studies, performed by different people, at different times and with different methods. Table 1 is not easy to understand, it should be simplify.

Lines 154-155: Linear regression should be used to assess the relationship between independent and dependent variables, not differences between groups.

Lines 161-162: you say that “The median right lateral VHS in the control and cardiac disease groups were 10.6 and 11.8, respectively (Table 1)” but in table 1 the same values are indicated as means.

Lines 167-168: Sensitivity, specificity, PPV and NPV should be reported together with their respective 95% confidence intervals.

Discussion and conclusions are not evaluable as materials and methods and results need to be completely revised

Author Response

Reviewer 3’s Report

This study could potentially provide clinicians important information on VHS values in healthy Brittanny Spaniels. However, the methods used to conduct the study are not adequate and the statistics are inappropriate. I suggest the authors to correct the weaknesses of their work and re-submit it to the journal in future.

Line 78: yo_u_ _s_a_y_ _t_h_a_t_ _“F_i_f_t_y_-t_h_r_e_e_ _o_w_n_e_d_ _a_n_i_m_a_l_s_ _w_e_r_e_ _r_e_c_r_u_i_t_e_d_ _r_e_t_r_o_s_p_e_c_t_i_v_e_l_y_”._ _H_o_w_e_v_e_r_,_ _i_n_ _l_i_n_e_ _1_2_1_ _y_o_u_ _s_a_y_ _“._._t_h_o_s_e_ _s_e_l_e_c_t_e_d_ _p_r_o_s_p_e_c_t_i_v_e_l_y_…” _Y_o_u_ _s_h_o_u_l_d_ _s_p_e_c_i_f_y_ _h_o_w_ _m_a_n_y_ _d_o_g_s_ _w_e_r_e_ _s_e_l_e_c_t_e_d_ _retrospectively and how many were included prospectively.

The authors thank the reviewer and apologize for this mistake. There is no prospectively selected group and we corrected this section.

Lines 80-81: I believe that the absence of murmur and the absence of clinical signs of heart failure are insufficient criteria to rule out cardiac disease. How were the ECG and echocardiography results of these dogs? How many of them have had an echocardiographic examination and how many have n_o_t_?_ _W_h_a_t_ _d_o_ _y_o_u_ _m_e_a_n_ _b_y_ _"_c_l_i_n_i_c_a_l_ _s_i_g_n_s_ _i_n_d_i_c_a_t_i_v_e_ _o_f_ _h_e_a_r_t_ _f_a_i_l_u_r_e_“._ _I_t_ _s_h_o_u_l_d_ _b_e_ _s_p_e_c_i_f_i_e_d_._ _M_o_r_e_o_v_e_r_,_ _also general physical examination, CBC, urinary analysis and biochemical panel are essential tools to assess the clinical status of the dogs and to rule out dehydration or other conditions that can affect heart size

The authors thank the reviewer and have precise this section as follow:” Dogs without a heart murmur and clinical signs indicative of heart failure were selected as a control group. Any dog with nonsinus arrhythmia, tachypnea, dyspnea, syncope, exercise intolerance, ascites, or cyanosis, The dogs that had an auscultable murmur and cardiac disease detected by echocardiography were selected as the cardiac disease group. The medical records of selected dogs were reviewed to extract the age at the radiographic examination, gender, body weight, body condition score (1 to 9) when available, clinical evaluation, and thoracic radiographs, and CBC, urinalysis and biochemical panel when available. In cases where echocardiography was performed, the results were also recorded. All the animals with a condition that can affect the size of the heart, such as dehydration or animals under perfusion erwe excluded from the study. »

Lines 85-86: how many dogs undergone echocardiography? it should be specified.

As precised in the result section, 53 dogs were recruited and undergone echocardiography. 43 of these dogs were included at the end  because of lack of quality of some of the radiographs.

Lines 91-92: W_h_a_t_ _d_o_e_s_ _i_t_ _m_e_a_n_ _“a_l_l_ _o_f_ _t_h_e_ _r_a_d_i_o_g_r_a_p_h_s_ _h_a_d_ _t_o_ _b_e_ _o_f_ _g_o_o_d_ _t_e_c_h_n_i_c_a_l_ _a_n_d_ _91 diagnostic q_u_a_l_i_t_y_ _“ _i_t_ _s_h_o_u_l_d_ _b_e_ _s_p_e_c_i_f_i_e_d_._

The authors thank the reviewer for his remark and have specified this part “To be included in the study, all of the radiographs had to be of good technical and diagnostic quality (density, contrast, sharpness). They were taken at the point of full inspiration, and care was taken to avoid any rotation of the body as this could influence the shape and the size of the cardiac silhouette [19].”

_

Lines 107-108: It should be specified who performed the echocardiographic examinations.

Lines 111-113: Usually, in small animals echocardiography, the Left Atrium to aortic ratio is performed in early ventricular diastole (Rishniw 2000, Hansson 2002). Why did you measure it at end diastole? Can you provide references about this method? What echocardiographic reference intervals did you consider normal for the left atrium and left ventricle?

To answer to these two interesting remarks, the authors improved the method section as follow: “All of the echocardiographic examinations were performed by the successive imaging assistants of the hospital, under supervision of the head of the imaging department (M.F.), two experienced cardiologists using an Esaote Mylab70XVG device. The operators were blinded to the radiographic results. Two-dimensional (2D), M-mode, and Doppler examinations were performed using standard views in unsedated dog. Specific measurements of interest included: the 2D end-diastolic left atrium to aortic ratio using the right parasternal short-axis view at the level of the heart base, and M-mode diastolic and systolic left ventricular internal diameters using the right parasternal short-axis in a transventricular view. All echocardiograms were performed on unsedated dogs in the standing position [20]. The following measurements were each taken over at least 3 cardiac cycles, and the mean was recorded as follows: the end-diastolic LA/Ao ratio obtained from the right parasternal short‐axis 2D [21], the left ventricular internal diameter at end‐diastole (LVIDd), and left ventricular internal diameter at end‐systole (LVIDs) measured on the M‐mode echocardiogram, obtained from the right parasternal short‐axis view [22,23]. M‐mode values were used to derive the fractional shortening (FS%). Normalized dimensions were calculated according to the following formulae: normalized LVIDd (LVIDDN) = LVIDd(cm)/(BW (kg)) 0.294; normalized LVIDs (LVIDSN) = LVIDs(cm)/(BW(kg)) [24]. Mitral valve disease was diagnosed when thickening and incomplete apposition of the valve leaflets during systole were observed, with secondary mitral valve regurgitation. The cardiac dogs were then classified into 2 groups according to the results of the echocardiography. All dogs with structural mitral valve disease with no echocardiographic evidence of cardiac remodeling formed the group 1. Dogs with more advanced mitral valve disease with echocardiographic findings of left atrial dilation with or without left ventricular dilation formed the group 2 [25]. “

Statistical analysis: Did you assess the distribution of the collected variables? The Student t-test is not suitable to compare more than two groups. Authors should use an adequate statistical test to compare more than two groups with each other. Linear regression should be used to assess the relationship between independent and dependent variables, not the correlation between two variables.

This section has been corrected as suggested by the reviewer “The statistical analyses were performed using a computerized statistical software package (R® software). For all of the VHS measurements, the mean and the standard deviation (SD) were calculated, and differences with p < 0.05 were considered significant. A non-parametric Wilcoxon-Mann-Whitney test was used to determine whether there were significant differences in the VHS values in right lateral recumbency in various subcategories. It was used to compare the VHS in the control group to the cardiac disease group with cardiac remodeling (group 2), and to compare cardiac diseased group 1 to group 2. Linear regression analysis was used to assess the correlation between the body score and the VHS. VHS of Brittany Spaniels with a body condition score 5 were compared to those with a body condition score  6 by mean of a Wilcoxon-Mann-Whitney test.

Lines 130-133:_ _t_h_e_ _m_e_a_s_u_r_e_ _o_f_ _c_e_n_t_r_a_l_ _t_e_n_d_e_n_c_y_ _“m_e_a_n_” _s_h_o_u_l_d_ _b_e_ _e_x_p_r_e_s_s_e_d_ _t_o_g_e_t_h_e_r_ _w_i_t_h_ _S_D_._ _

This has been corrected in the text : The mean age in our sample was 9.2 +/-  4,17 years (2 to 17), 7.9 +/- 3,79 years in the healthy group (5.5 to 11.5) and 11.9 +/- 3,65 in the cardiac disease group (8.2 to 15.6). The mean weight was 16.4 kg+/- 3,73 (10 to 24), the body condition score was available for 26 dogs, with a mean of 5.9 +/- 3,1 (4 to 8).

Lines 140-142: What criteria did you use to diagnose mitral valve disease? You should specify this in M&M 2 What reference intervals did you use to diagnose left atrial enlargement? You should specify this in M&M

What reference intervals did you use to diagnose left ventricular enlargement? You should specify this in M&M

As precised below, All echocardiograms were performed on unsedated dogs in the standing position [20]. The following measurements were each taken over at least 3 cardiac cycles, and the mean was recorded as follows: the end-diastolic LA/Ao ratio obtained from the right parasternal short‐axis 2D [21], the left ventricular internal diameter at end‐diastole (LVIDd), and left ventricular internal diameter at end‐systole (LVIDs) measured on the M‐mode echocardiogram, obtained from the right parasternal short‐axis view [22,23]. M‐mode values were used to derive the fractional shortening (FS%). Normalized dimensions were calculated according to the following formulae: normalized LVIDd (LVIDDN) = LVIDd(cm)/(BW (kg)) 0.294; normalized LVIDs (LVIDSN) = LVIDs(cm)/(BW(kg)) [24]. Mitral valve disease was diagnosed when thickening and incomplete apposition of the valve leaflets during systole were observed, with secondary mitral valve regurgitation. The cardiac dogs were then classified into 2 groups according to the results of the echocardiography. All dogs with structural mitral valve disease with no echocardiographic evidence of cardiac remodeling formed the group 1. Dogs with more advanced mitral valve disease with echocardiographic findings of left atrial dilation with or without left ventricular dilation formed the group 2 [25]. “

Lines 143-147: I believe that it is not correct to statistically compare the results obtained from two different studies, performed by different people, at different times and with different methods. Table 1 is not easy to understand, it should be simplify.

Table 1 has been removed as suggested by reviewer 1.

Statistical comparison between our study and Buchanan’s has been removed.

Lines 154-155: Linear regression should be used to assess the relationship between independent and dependent variables, not differences between groups.

Linear regression has been used to assess the correlation between the body score and the VHS that are independent variable.

Lines 161-162: y_o_u_ _s_a_y_ _t_h_a_t_ _“T_h_e_ _m_e_d_i_a_n_ _r_i_g_h_t_ _l_a_t_e_r_a_l_ _V_H_S_ _i_n_ _t_h_e_ _c_o_n_t_r_o_l_ _a_n_d_ _c_a_r_d_i_a_c_ _d_i_s_e_a_s_e_ _g_r_o_u_p_s_ _w_e_r_e_ _1_0_._6_ _a_n_d_ _1_1_._8_,_ _r_e_s_p_e_c_t_i_v_e_l_y_ _(_T_a_b_l_e_ _1_)_” _b_u_t_ _i_n_ _t_a_b_l_e_ _1_ _t_h_e_ _s_a_m_e_ _v_a_l_u_e_s_ _a_r_e_ _i_n_d_i_c_a_t_e_d_ _a_s_ _m_e_a_n_s_._ _

As suggested by the reviewer, table 1 has been removed. The values are effectively mean and not median. This has been checked in the text.

Lines 167-168: Sensitivity, specificity, PPV and NPV should be reported together with their respective 95% confidence intervals.

The authors totally agree with the reviewer’s comment and the sentence has been corrected.” With a threshold value of 11.1 vertebrae, the sensitivity, specificity, and positive and negative predictive values were 90 % (62-98), 72 % (55-84), 53 % (38-66) and 96 % (90-100), respectively. The Youden’s index is estimated at 0,63. »

Round 2

Reviewer 1 Report

The author have corrected all the suggestion, so I think the article could be published.

Reviewer 2 Report

In the current version according to the questions essential changes were done. The groups are now correctly classified, and the comparisons are now correct. From my point of view all questions are solved and I would think this study should now be published. Thanks to the authors for their great work.

Reviewer 3 Report

I thank the authors for making most of the required changes to the manuscript. However, there are still important methodological errors:

  • The most important problem is the statistical analysis: The test used are not adequate for the objectives that the authors want to pursue, fort this reason the results cannot be evaluated.
  • Authors did not specify which cut-off values of LA/Ao, LVIDdN and LVIDsN they consider to diagnose cardiac remodeling and to classify dogs in subgroups 1 or 2. For this reason the results about differences within groups cannot be evaluated.

I suggest that the authors ask for the assistance of a statistician to solve problems and after that resubmit the manuscript.